# Genome-Wide Identification, Classification, Expression Analysis, and Screening of Drought and Heat Resistance-Related Candidates of the *Rboh* Gene Family in Wheat

**DOI:** 10.3390/plants13233377

**Published:** 2024-11-30

**Authors:** Miyuan Cao, Yue Zhang, Xiaoxiao Zou, Huangping Yin, Yan Yin, Zeqi Li, Wenjun Xiao, Shucan Liu, Yongliang Li, Xinhong Guo

**Affiliations:** 1College of Biology, Hunan University, Changsha 410082, China; caomiyuan@hnu.edu.cn (M.C.); zhangyue_one@hnu.edu.cn (Y.Z.); xxzou@hnu.edu.cn (X.Z.); yinhuangping@hnu.edu.cn (H.Y.); yinyan91@foxmail.com (Y.Y.); lizeqi2023@hnu.edu.cn (Z.L.); xiaowj90@hnu.edu.cn (W.X.); liushucan@hnu.edu.cn (S.L.); 2Chongqing Research Institute, Hunan University, Chongqing 401120, China; 3Northeast Institute of Geography and Agroecology, Chinese Academy of Sciences, Changchun 130102, China

**Keywords:** *Triticum aestivum*, *Rboh*s, abiotic stress, reactive oxygen species, expression patterns

## Abstract

Plant respiratory burst oxidase homologs (Rbohs) are key enzymes that produce reactive oxygen species (ROS), which serve as signaling molecules regulating plant growth and stress responses. In this study, 39 *TaRboh* genes (*TaRboh01*–*TaRboh39*) were identified. These genes were distributed unevenly among the wheat genome’s fourteen chromosomes, with the exception of homoeologous group 2 and 7 and chromosomes 4A, as well as one unidentified linkage group (Un). TaRbohs were classified into ten distinct clades, each sharing similar motif compositions and gene structures. The promoter regions of *TaRboh*s contained *cis*-elements related to hormones, growth and development, and stresses. Furthermore, five *TaRboh* genes (*TaRboh26*, *TaRboh27*, *TaRboh31*, *TaRboh32*, and *TaRboh34*) exhibited strong evolutionary conservation. Additionally, a Ka/Ks analysis confirmed that purifying selection was the predominant force driving the evolution of these genes. Expression profiling and qPCR results further indicated differential expression patterns of *TaRboh* genes between heat and drought stresses. *TaRboh11*, *TaRboh20*, *TaRboh22*, *TaRboh24*, *TaRboh29*, and *TaRboh34* were significantly upregulated under multiple stress conditions, whereas *TaRboh30* was only elevated in response to drought stress. Collectively, our findings provide a systematic analysis of the wheat *Rboh* gene family and establish a theoretical framework for our future research on the role of *Rboh* genes in response to heat and drought stress.

## 1. Introduction

Respiratory burst oxidase homologs (Rbohs) are known as NADPH oxidases. They are pivotal in ROS production, which is the main source of these reactive molecules in plants [1,2]. Plant Rbohs, homologous to the gp91^phox^ catalytic subunit in mammalian phagocytic cells, are plasma membrane proteins featuring six conserved transmembrane helices, two heme groups, a C-terminal domain with FAD and NADPH, and two N-terminal EF-hand domains for calcium binding [3]. These structural features enable Rbohs to catalyze the production of ROS by using NADPH or NADH as electron donors, leading to the extracellular conversion of O_2_ into superoxide (O_2_^–^), which is then converted into hydrogen peroxide (H_2_O_2_) by superoxide dismutase (SOD) [3,4]. The ROS generated by *Rboh*s act as key signaling molecules, regulating plant growth, development, and stress responses, underscoring the central role of *Rboh*s in plant regulation [2,3].

The first *Rboh* family gene in plants, *OsRbohA*, was identified in *Oryza sativa* L. in 1996 [1]. Subsequently, a large number of *Rboh* genes have been identified in different plant species. They included 10 in *Arabidopsis thaliana* (*AtRbohA*
− *AtRbohJ*) [2], 20 in *Nicotiana Tabacum* (*NtabRbohA* − *NtabRbohT*) [5], 9 in rice (*OsRbohA* − *OsRbohI*) [6], 27 in *Gossypium barbadense* (*GbRboh1* − *GbRboh27*) [7], and 15 in *Zea mays* (*ZmRBOH1*~ *ZmRBOH15*) [8]. *Rboh* genes exhibit diverse tissue-specific transcription levels, providing insights into their roles in plant biology [9]. For example, three of those in *Arabidopsis*, (*AtRbohA, AtRbohG*, and *AtRbohI*) were expressed in roots, two (*AtRbohH* and *AtRbohJ*) were specifically expressed in pollen, and another two (*AtRbohD* and *AtRbohF*) were expressed in all tissues [2]. For those in rice, all but two (*OsRbohD* and *OsRbohI*) of the *OsRboh* genes were expressed in roots, leaves, shoots, and calli [10]. Additionally, plant *Rboh genes* are involved in various stress responses, including drought and heat. Overexpression of the *AtRbohI* gene in *Arabidopsis* enhanced the drought stress response [11]. *AtRbohB* and *AtRbohD* mediated signal transduction under heat stress, with *AtRbohD* also responded to drought stress [12]. *OsRbohA* participated in complex signal transduction processes and contributed to plant growth and drought resistance [6]. *OsRbohB* mediated ROS and ABA signaling in response to drought stress [13]. Knocking down *ZmRBOHC* reduced the response of maize to drought stress [14]. *SlRbohB* positively regulated the drought tolerance in tomato [15].

Bread wheat (*Triticum aestivum* L.) is one of the third-largest cereal crops globally [16]. Recognizing and studying the *Rboh* gene family in wheat is essential to ensuring food security and increasing productivity, since abiotic stresses restrict plant growth and present significant challenges to the world’s food supply. However, current research on the *Rboh* gene family in wheat either encounters flaws in the identification process [17] or employs improper selection criteria for *Rboh* genes [18], leading to improper identification of *Rboh* members. In this study, we used bioinformatics techniques to identify the *Rboh* gene family in wheat, determine their chromosomal locations, evolutionary relationships, conserved motifs, structures, cis-elements, collinearity, protein interaction networks, and miRNA targets. Additionally, we analyzed the expression profiles of *TaRboh* genes under different conditions using RNA-Seq data [19] and validated these profiles with quantitative real-time PCR (qRT-PCR) experiments. Overall, our findings will assist in elucidating the molecular mechanisms of *TaRboh* genes in responses to drought, heat, and combinations of these stress conditions in wheat.

## 2. Results

### 2.1. Genome-Wide Identification, Chromosome Distribution, and Physicochemical Properties of Rboh Gene Family Members in Wheat

Using HMMER 3.0 software and conserved domain identification, 39 *TaRboh* genes encoding putative proteins with conserved NADPH_Ox, Ferric_reduct, FAD_binding_8, and NAD_binding_6 structural domains were identified using a Hidden Markov Model (HMM)-based approach combined with SMART structural domain validation. Among them, TraesCS1B02G295200 and TraesCS1D02G284900 contained three and four EF-hand motifs, respectively; twenty-eight Rboh proteins had two EF-hand motifs, eight Rboh proteins had only one EF-hand motif, and one Rboh protein (TraesCS3B02G212900) lacked an EF-hand motif (Appendix A). Based on their genomic localization in wheat and widely accepted standard of gene nomenclature, the 39 *TaRboh* genes were designated *TaRboh01* to *TaRboh39*. The *TaRboh* genes were unevenly distributed across chromosomes 1A, 1B, 1D, 3A, 3B, 3D, 4B, 4D, 5A, 5B, 5D, 6A, 6B, and 6D. Furthermore, most *TaRboh* genes were located in the terminal regions of the chromosomes, with few in the central regions (Figure 1). Analysis of the physicochemical properties of TaRboh proteins (Appendix A) revealed that the predicted Rboh proteins comprised between 800 (TaRboh02) and 1223 (TaRboh04) amino acids, with molecular weights ranging from 90.83 kDa (TaRboh02) to 139.42 kDa (TaRboh02). Their pIs varied from 8.88 (TaRboh09) to 9.47 (TaRboh18), indicating that most of the TaRboh proteins were basic. For the instability index (II), only six Rboh proteins (TaRboh06-TaRboh07, TaRboh12, TaRboh36-TaRboh38) were predicted to be stable, with instability index values below 40. The GRAVY value was less than zero, suggesting that these proteins were hydrophilic. Moreover, subcellular localization prediction revealed that all of these TaRboh proteins were localized to the cell membrane (Appendix A).

### 2.2. Sequence Analysis, Phylogenetic Analysis, and Classification of Rboh Members

Thirty-nine TaRboh protein sequences from wheat, ten AtRboh protein sequences from *Arabidopsis*, and nine OsRboh protein sequences from rice were used in a phylogenetic analysis using MEGA 11.0 software to understand the evolutionary relationships of the *TaRboh* gene family in wheat and other species. As shown in the phylogenetic tree obtained, the TaRbohs were dispersed among the five groups into which the Rbohs were clustered (Figure 2A). Twelve TaRboh proteins were in group A, nine in group B, six in group C, six in group D, and six in group E (Figure 2B). Subsequent investigation showed that the Rboh proteins from rice and wheat were more closely grouped, suggesting a closer genetic relationship between the TaRboh family members between these two species.

### 2.3. Structure and Conserved Motifs of TaRbohs

To explore the diversity of *TaRboh* genes, in silico phylogenetic tree construction, gene structure analysis, and motif identification were performed (Figure 3). The 39 *TaRboh* genes formed 10 groups based on the analysis using MEGA 11.0 software. Groups VI, VII, and X had the most *TaRboh* gene members (six each), while the other groups contained two or three *TaRboh* genes each (Figure 3A). Additionally, the gene structure analysis (Figure 3C) revealed that most of the *TaRboh*s had untranslated regions (UTRs). The exceptions included *TaRboh01*, *04*, *09*, *28*, and *36*. All but three (*TaRboh28*, *33*, and *36*) of these *TaRboh*s possessed introns. *TaRboh04* has the highest number of exons (16), while the numbers of exons possessed by other *TaRboh* genes varied between 1 and 14. As shown in Figure 3B, different groups of *TaRboh* genes had different motif compositions, but motifs for these genes within the same group are highly preserved. All except Group II genes had 15 motifs. Those in Group II lacked motif 14. According to domain alignment, the NADPH_Ox domain was made up of motifs 13 and 15, the NAD_binding_6 domain was made up of motifs 3, 5, 8, and 9, the Ferric_reduct domain was made up of motifs 2, 7, and 11, and the FAD_binding_8 domain was made up of motifs 1. In general, *TaRboh*s with the same types and arrangements of motifs were clustered into the same group. Genes on adjacent branches of the phylogenetic tree with similar numbers and arrangements of motifs and gene structures may have similar biological functions.

### 2.4. Cis-Elements in TaRboh Genes

*Cis*-elements in promoter regions specifically bind to transcription factors, thereby regulating the expression of downstream genes [20]. The *cis*-elements in the promoters of the *TaRboh* genes were examined using the PlantCARE website. A total of 3862 *cis*-elements were found in promoters of the 39 *TaRboh* genes. They comprised 1192 binding site elements, 197 elements connected to plant growth and development, 504 phytohormone responsive elements, 1117 elements related to abiotic and biotic stressors, and 852 elements without a functional description (Appendix A). As shown in Figure 4, the majority of the biotic and abiotic stress elements are linked to wounding (CTAG-motif, TC-rich repeats, WRE3, and WUN-motif), dehydration (DRE/CRT-core and DRE1), elicitor responses (W box), low temperature (LTR), light (A-box, ACE, AE-box, ATCT-motif, Box 4, Box II-like sequence, chs-CMA2a, chs-Unit 1 m1, Gap-box, GATA-motif, G-Box, GT1-motif, GTGGC-motif, I-box, Pc-CMA2a, Pc-CMA2c, Sp1, MRE, TCCC-motif, and TCT-motif), anoxic specific inducibility (GC-motif), anaerobic induction (ARE), replication (E2Fb), stress response (STRE), and drought stress (MYB, MYB-binding site, MYB-like sequence, and MYC). Phytohormone-responsive elements included those responsive to auxin (AuxRR-core, TGA-box, and TGA-element), gibberellin (GARE-motif, P-box, and TATC-box), salicylic acid (as-1 and TCA-element), abscisic acid (ABRE, ABRE3a, and ABRE4), estrogen (ERE), glucocorticoid (GRA), and MeJA (CGTCA-motif and TGACG-motif). Plant growth and development elements are associated with endosperm expression (AAGAA-motif and GCN4_motif), xylem-specific expression (AC-I and AC-II), meristem expression (CAT-box), tissue activation (CCGTCC motif and CCGTCC-box), circadian control (circadian), meristem specific activation (dOCT), root-specific (motif I), zein metabolism (O2-site), cell cycle (re2f-1 and telo-box), seed-specific regulation (RY-element), and DNA regulation (Y-box). Abiotic and biotic stress elements, phytohormone-responsive elements, and plant growth and development elements are the three categories of *cis*-elements found in the promoters of each *TaRboh* gene; *TaRboh20*, *TaRboh12*, and *TaRboh13* had the largest numbers of these elements.

### 2.5. Collinearity Relationship of the TaRboh Gene Family

Gene tandem repeats and fragment repeats are essential for gene family expansion in plants [21]. Forty-eight pairs of identical *TaRboh* genes were found using in silico collinearity analysis in wheat; these genes are unevenly distributed on homoeologous group 1, 5, and 6, as well as chromosomes 4B and 4D (Figure 5). Segmental duplications were found in all of the 48 gene pairs using MCScanX. However, tandem duplications were found in none of them. These findings imply that segmental duplications have played a major role in the evolution and diversification of the *TaRboh* genes. Values of nonsynonymous (Ka) and synonymous (Ks) ratios were computed in order to examine the underlying evolutionary selection pressure on the *TaRboh* gene family. Ka/Ks ratios for all of the 48 homologous gene pairs were smaller than 1 (Appendix A), indicating that strong purifying selection might have been experienced by these genes during the evolutionary process.

As shown in Figure 6A, estimated divergence times between wheat and the other six species (*Brachypodium distachyon*, rice, maize, *Glycine max*, *Arabidopsis*, and *Aegilops tauschii*) were 3.54 Mya, 33.0 Mya, 47.0 Mya, 49.1 Mya, 108.0 Mya, and 159.6 Mya, respectively. Based on the phylogenetic tree with wheat as the reference genome, 44 homologous pairs were found between wheat and *Aegilops* and they included 33 *Rboh* genes in wheat and 12 genes in *Aegilops*. From the twelve genes in *Aegilops*, eleven homologous pairs were identified between *Aegilops* and *Brachypodium* and they included six genes in *Aegilops* and seven in *Brachypodium*. Furthermore, twelve homologous pairs were identified between the seven genes in *Brachypodium* and seven genes in rice. Additionally, eleven homologous pairs were identified between seven genes in rice and six genes in maize. Three homologous pairs were identified between two genes in maize and two genes in *Glycine*. One homologous pair was identified between two genes in *Glycine* and two genes in *Arabidopsis* (Figure 6B). Among these, five *TaRboh* genes (*TaRboh26*, *TaRboh27*, *TaRboh31*, *TaRboh32*, and *TaRboh34*) were involved in twenty-four continuous collinear pairs. All of the computed Ka∕Ks values for these collinear pairs (Appendix A) were less than 1.0, consistent with the likelihood that purifying selection is a main drive force during evolution of these genes. Five of these *TaRboh* genes (*TaRboh26*, *TaRboh27*, *TaRboh31*, *TaRboh32*, and *TaRboh34*) also showed strong evolutionary conservation, and they seemed to be important for plant adaptation and evolution.

### 2.6. Protein–Protein Interactions and miRNA Targeting Events in TaRboh Genes

Protein complexes, which include a great deal of protein interaction, are necessary for the majority of life activities in an organism. To comprehend complicated biological processes, it is imperative to identify these interactions [22]. Potential interactions between TaRboh proteins were predicted using the STRING database. While TaRboh19 was found to have no interactions, the 38 TaRboh proteins were found to have 312 interaction pairs altogether. Of the TaRboh proteins, 13 (TaRboh03, TaRboh07, TaRboh12, TaRboh14, TaRboh17, TaRboh20, TaRboh22, TaRboh27, TaRboh32, TaRboh35, TaRboh37, TaRboh38, and TaRboh39) were predicted to be central nodes in the network based on the degree of connectivity within the network. Each of these proteins had 26 connections with other genes (Figure 7A).

MicroRNAs (miRNAs) are a class of endogenous, small RNAs with approximately 22 nucleotides in length. Plant miRNAs are important for growth, development, and stress responses. They also take part in post-transcriptional gene regulation [23,24]. The psRNATarget service was utilized to investigate the ways in which miRNAs control *TaRboh* gene expression. Of the *TaRboh* genes, 34 were shown to be putative targets of 31 different miRNAs. Tae-miR408 regulated the largest number of *TaRboh* genes (*TaRboh04*, *TaRboh14*, *TaRboh20*, *TaRboh22*, *TaRboh27*, *TaRboh35*, *TaRboh37*, *TaRboh38*, and *TaRboh39*). Furthermore, *TaRboh16* was the primary target (Figure 7B).

### 2.7. Expression of TaRboh Genes in Different Stress Conditions

Drought and heat stress are the main factors limiting plant growth and reproduction, and they often occur simultaneously [25]. To further investigate the potential role of *TaRboh* genes, wheat was grown under a variety of stress conditions, including drought stress for one hour (DS-1), drought stress for six hours (DS-1), heat stress for one hour (HS-1), heat stress for six hours (HS-6), combined heat and drought stress for one hour (DHS-1), and combined drought and heat stress for six hours (DHS-6). An analysis of RNA-seq transcriptomic data was performed to evaluate the expression profiles of the *TaRboh* genes in various stress conditions using the Wheat Expression Browser (expVIP) (www.wheat-expression.com (accessed on 7 December 2023)). This analysis showed the distinct expression of these genes under different stress scenarios (Figure 8A). Compared to those in the controls, three-quarters of the *TaRboh* genes had higher expression at DS-1 and lower expression at DS-6. Although the expression of 11 *TaRboh* genes was not significantly altered at HS-1, substantial upregulation for 12 *TaRboh* genes was detected at DHS-6. Additionally, during drought stress, expressions of some genes (*TaRboh14*, *TaRboh17*, and *TaRboh20*) did not significantly change. However, the expressions of these genes were significantly increased under heat stress and combined heat and drought stress. *TaRboh28*, *TaRboh33*, and *TaRboh36* showed no significant change under other conditions, but had significantly increased expression under drought stress.

To further verify the reliability of the transcriptome data, nine representative members from the thirty-nine *TaRboh* genes were selected for quantitative real-time PCR (qRT-PCR) experiments. As shown in Figure 8B, nine different gene expression profiles show different degrees of responsiveness to heat, drought, and a combination drought and heat conditions. Among these, six genes (*TaRboh09*, *TaRboh11*, *TaRboh22*, *TaRboh24*, *TaRboh30*, and *TaRboh34*) show expression patterns that align with the RNA-seq findings. Nevertheless, the expression profiles of three genes (*TaRboh09*, *TaRboh23*, and *TaRboh29*) differ from the RNA-seq results. The RNA-seq transcriptome data indicate that *TaRboh20* is downregulated at HS-6 compared to the control group, the expression of *TaRboh29* at HS-6 is lower than at DHS-6, and *TaRboh23* is upregulated at DHS-1 (Figure 8A). In contrast, the qRT-PCR results show that *TaRboh20* is upregulated at HS-6, *TaRboh23* exhibits a significant increase in expression at DHS-1, and *TaRboh29* shows significantly higher expression levels at HS-6 than at DHS-6. These findings suggest that distinct *TaRboh* genes have distinct patterns of expression in response to heat, drought, and combined heat and drought stress.

## 3. Discussion

*Rboh* genes encode Rboh enzymes in plants. These enzymes are essential for generating ROS and controlling plant shape, growth, development, and responses to biotic and abiotic stresses [26]. *Rboh* genes have been thoroughly investigated in a wide range of plant species. Hu et al. [17] and Sharma et al. have both attempted to identify *Rboh* family genes in wheat, but their findings are flawed. Hu’s 2018 study reported 46 *Rboh* genes. However, it did not mandate the presence of all four domains (NADPH_Ox, Ferric_reduct, FAD_binding_8, and NAD_binding_6) for classification as *Rboh* genes. Furthermore, the wheat genome that was employed in their study is now only marginally relevant due to advances in sequencing technology. Sharma’s recent research in wheat revealed 40 *TaRboh* genes [18]. With the exception of the extra gene *TraesCS1B02G295100* that is included in Sharma’s investigation, all 39 of the *TaRboh* genes that we found through in silico analysis (Figure 1) are included in his findings. Domain analysis verified the presence of the NADPH_Ox, Ferric_reduct, FAD_binding_8, and NAD_binding_6 domains in the 39 TaRboh proteins that we discovered (Appendix A). TraesCS1B02G295100, on the other hand, lacks the NADPH_Ox domain and only has the Ferric_reduct, FAD_binding_8, and NAD_binding_6 domains. The NADPH_Ox domain, located at the EF hand’s N-terminus, is crucial for generating ROS via Ca2+ signaling that is mediated by Rboh proteins [27]. Considering that the NADPH_Ox domain is a crucial component of Rboh genes, TraesCS1B02G295100 most likely functions as a ferric reduction oxidase (FRO) protein as opposed to a Rboh protein [28]. Therefore, we hypothesize that *TraesCS1B02G295100* is a *FRO* gene rather than a *TaRboh* gene.

Notably, the number of *Rboh* genes in wheat is significantly higher than those in *Arabidopsis* (10) [2], rice (9) [6], maize (15) [8], *Aquilaria* (14) [29], and eggplant (8) [30], yet the number of *Rboh* genes does not seem to correlate with genome sizes [31]. Additionally, Chromosomes in homoeologous groups have an equal amount of *TaRboh* genes located in consistent positions on each chromosome. However, the distribution of *TaRboh* genes is not uniform in homoeologous groups 1 and 4, where certain genes are restricted to specific chromosomes (Figure 1). Whole-genome duplication (WGD) is the most likely explanation for these observations, as it caused the allohexaploid nature (AABBDD) of common wheat. This explains the wheat’s enormous genome size and the increased number of members of the *TaRboh* gene family, which are frequently seen as three homologous alleles. But as polyploid systems get closer to states that resemble diploid states, gene loss could happen as a result of copy number reduction, pseudogenization, or silencing. Consequently, only particular chromosomes within homologous groups 1 and 4 contain *TaRboh* homologous genes [32,33,34,35]. Furthermore, collinearity analysis in wheat revealed 48 pairs of identical genes, all of which are segmental duplications (Figure 5). This finding highlights the significance of segmental duplication in the evolution of the *TaRboh* gene family. Thus, these results suggest that both segmental and whole-genome duplications may play key roles in the evolution and expansion of the *TaRboh* gene family in wheat.

Rboh protein sequences from wheat, rice, and *Arabidopsis* were used to create phylogenetic trees. Similarly to those reported earlier [29], the *Rboh* genes were clustered into five separate groups (Figure 2). Compared with those in *Arabidopsis*, *Rboh*s in wheat have closer homologous links with those in rice, indicating that these genes probably share similar structures and functions within the same group of the evolutionary tree. Group D comprises *TaRboh03*, *TaRboh07*, *TaRboh12*, *TaRboh14*, *TaRboh17*, and *TaRboh20* in wheat, and *OsRbohA* in rice, which is known for its role in drought resistance [6]. These wheat genes belong to the same group based on phylogenetic, motif, and gene structural analyses (Figure 3). They all have *cis*-regulatory elements (such as MYB and MYC) that are associated with drought tolerance (Figure 4). Results from the transcriptome analysis indicated that *TaRboh03*, *TaRboh07*, *TaRboh12*, *TaRboh14*, *TaRboh17*, and *TaRboh20* were related to drought-tolerance. Results from RT-qPCR analysis confirmed that *TaRboh20* was upregulated under drought stress (Figure 8). The other five genes also seem to be related to drought stress, although experiments are needed to validate this possibility.

In comparative genomics, homology inference is essential. However, heterozygosity, polyploidy, WGD, and repetitive sequences may limit the effectiveness of approaches based solely on sequence similarity and obscure homologous relationships [36,37]. Compared to other variables, gene content and gene order have been found to be more stable during plant evolution, making collinearity analysis especially valuable for homology inference [38]. It also aids in illuminating ancestral polyploid events [39]. We showed in this study that, although the homologous relationships of several *TaRboh* genes have gradually diminished, only five *TaRboh* genes (*TaRboh26*, *TaRboh27*, *TaRboh31*, *TaRboh32*, and *TaRboh34*) have maintained homologous relationships throughout evolution (Appendix A). This may be attributed to the larger genome of wheat and the increased gene count resulting from WGD, which far exceed those of *Arabidopsis*, ultimately leading to the loss of most duplicated genes [40]. These findings indicate that *TaRboh26*, *TaRboh27*, *TaRboh31*, *TaRboh32*, and *TaRboh34* are highly conserved throughout evolution and have significant research value. However, further investigation through experiments is needed.

Transcription factors (TFs) and microRNAs (miRNAs) are important regulators of gene expression under stress that lessen the effects of adverse environmental conditions in plants [41]. The position, timing, and degree of gene expression are all determined by *cis*-elements, which are individual TF-binding sites [42]. MiRNAs attach to target mRNAs at precise locations and cleave them post-transcriptionally to control gene expression and prevent protein translation [43]. As shown in Figure 4, *cis*-element analyses of *TaRboh* gene promoters reveal a variety of *cis*-elements which could be categorized into three functional groups: stress response, hormone response, and developmental regulation. Notably, 61.4% of these elements are associated with biotic and abiotic stress responses. They included DRE core, DRE1, MYB, and MYC elements related to drought stress. Additionally, ABRE, a hormone-responsive element involved in ABA-responsive gene expression, plays a significant role in drought stress response [44]. Therefore, *TaRboh* genes are likely involved in drought stress responses. Furthermore, we identified 31 distinct miRNAs and predicted interactions between them and the *TaRboh* genes in wheat (Figure 7B). Tae-miR408 was found to have the most targets among the *Rboh* genes, and it is known to be involved in regulating heading time, phosphorus deprivation, high salinity, heavy copper stress, and stripe rust stress [45,46,47]. Our results indicate that Tae-miR408 may target *TaRboh04*, *TaRboh14*, *TaRboh20*, *TaRboh22*, *TaRboh27*, *TaRboh35*, *TaRboh38*, and *TaRboh39*, which requires experimental validation.

Drought and heat are major abiotic stressors that limit wheat production and often occur together. This multi-stress scenario affects plants in ways that cannot be predicted solely based on the impact of individual stresses [48,49]. Current research indicates that plant *Rboh* is capable of responding to heat, drought, and combined drought and heat stress. Through the *OsRALF45*/*46*-*OsMRLK63*-*OsRboh*s signaling circuit, OsMRLK63 (a kinase) controls intracellular ROS levels, hence regulating drought tolerance [50]. The FERONIA Receptor-Like Kinase in tomatoes was used by the BZR1 transcription factor to facilitate ROS signaling from *RBOH1* in response to heat stress; the *bzr1* mutant downregulated *RBOH1* expression, which reduced drought resistance, which could be partially restored by exogenous H_2_O_2_ [51]. *ERF74* and *ERF75* are functionally redundant in *Arabidopsis*; overexpression of *ERF74* increased ROS levels and improved resistance to heat and drought by upregulating *RbohD*, whereas the *erf74* mutant and *erf74*; *erf75* double mutant have decreased stress resistance and *RbohD* expression [52]. Additionally, hormones such as ABA (abscisic acid) and JA (jasmonic acid) also impact wheat drought resistance [53]. Expression profiling and quantitative data obtained from this study confirmed that *TaRboh* genes respond to drought, heat, and combined drought and heat stresses to varying degrees. Furthermore, several *TaRboh* genes that are elevated under single stress (heat or drought) conditions apparently have their increased expression inhibited by the dual stress (heat and drought). For instance, according to transcriptome data and qPCR results, *TaRboh22*, *TaRboh24*, and *TaRboh29* expression levels were significantly greater at HS-6 than at DHS-1 and DHS-6, and *TaRboh30* expression was significantly higher at DS-6 than at DHS-1 and DHS-6 (Figure 8). More experimental research is needed to understand this phenomenon. *TaRboh22*, a homologue of *AtRbohD*, was likely involved in responses to various abiotic stresses including drought, heat, and their combination. *TaRboh30*, with high sequence similarity to *OsRbohA*, also responds to drought stress. We found it interesting that under different stress circumstances, duplicated genes show comparable expression patterns. Examples of genes that exhibit functional redundancy among these duplicated genes include *TaRboh22*, *TaRboh24*, and *TaRboh29*. They all exhibited increased expression in response to heat, drought, and combined stress. The most notable expression increase occurred at HS-6. In summary, distinct *TaRboh*s expression patterns were noted. A possible explanation for the discrepancies between the qRT-PCR and RNA-seq results for the three *TaRboh* genes (*TaRboh09*, *TaRboh23*, and *TaRboh29*) is that we performed the treatment when the wheat reached the three-leaf stage, whereas the reference literature for RNA-seq [19] did not specify the exact timing of the treatment. In the subsequent in-depth study of the functions of these three *TaRboh* genes (*TaRboh09*, *TaRboh23*, and *TaRboh29*), we will conduct expression analysis on materials at different growth stages and under different stress treatments. The variables impacting the uneven expression patterns of *TaRboh* genes, such as the location, timing, and length of stress treatment, may be further analyzed by more research. Furthermore, a thorough investigation of the possible mechanisms behind the effects of heat and drought on wheat development is warranted.

## 4. Materials and Methods

### 4.1. Plant Materials and Stress Treatments

Wheat seeds (*Triticum aestivum* L. cv. Fielder) [54] were germinated on moist filter paper at 4 °C for 5 days, followed by 5 days at 12 °C. The seedlings were then transferred to a greenhouse with the setting of light 16 h/22 °C and dark 8 h/16 °C. The plants were grown until the three-leaf stage for stress treatments. The plants were subjected to PEG-induced dehydration mimicking drought stress (25% PEG6000 solutions), heat stress (37 °C incubator), and combined PEG-induced dehydration mimicking drought and heat stress (37 °C incubator and 25% PEG6000 solutions). Samples were collected at 0, 1, and 6 h post-treatment and were used to determine the expression levels of nine selected *TaRboh* genes under different conditions and time points.

### 4.2. Bioinfomatics Identification of TaRboh Family Genes in Triticum Aestivum L.

To identify *Rboh* genes in wheat, a sequence alignment was performed using HMMER 3.0 software (E-value < 1.0 × 10^−15^). The Hidden Markov Model (HMM) profile of the respiratory burst NADPH oxidase domain (PF08414; PF01794; PF08022; PF08030) was queried from the Pfam database (available online: pfam-legacy.xfam.org/ (accessed on 7 Decmeber 2023)). Wheat gene sequences, protein sequences, cDNA files, and gene annotation files were obtained from the Ensembl Plants database (https://plants.ensembl.org/index.html, accessed on 14 November 2023). Protein sequences of candidate Rbohs were further screened using SMART (http://smart.embl-heidelberg.de/, accessed on 7 Decmeber 2023) for domain prediction. Only those sequences containing the NADPH_Ox, Ferric_reduct, FAD_binding_8, and NAD_binding_6 domains were considered to be *Rboh* genes. Chromosomal localization and visualization of *TaRboh*s were performed using TBtools V2.136 software [55]. Additionally, the number of amino acids, molecular weight, theoretical pI, instability index, aliphatic index, and grand average of hydropathicity of the *TaRboh* gene family were obtained from the ExPASy website (http://web.expasy.org/protparam/, accessed on 7 Decmeber 2023) [56]. In silico Subcellular localization was predicted using the Plant-mPLoc website (version 2.0) [57] (http://www.csbio.sjtu.edu.cn/bioinf/plant-multi/, accessed on 7 Decmeber 2023).

### 4.3. Multiple Alignments and Phylogenetic Tree Analysis

Protein sequences of TaRbohs were aligned using the Clustal X 2.1 tool [58], and the alignments were edited using the ESPrit 2.2-ENDscript 1.0 tool [59]. *Arabidopsis* and rice sequence data used in this study were obtained from the National Center for Biotechnology Information (NCBI) (http://www.ncbi.nlm.nih.gov). A phylogenetic tree was constructed using the MEGA 11.0 software with the neighbor-joining method, and reliability was confirmed using 1000 replicates of the bootstrap analysis [60]. Additionally, iTOL 6.9 (https://itol.embl.de) [61] was used to visualize the phylogenetic tree.

### 4.4. Conserved Motif and Gene Structure Analysis of TaRbohs

The MEME server v 5.5.5 (http://meme-suite.org/tools/meme, accessed on 7 Decmeber 2023) [62] and the Gene Structure Display Server (GSDS) website (http://gsds.gao-lab.org/) were used for the analysis of conserved motifs and gene structure of *TaRboh*s, respectively. For the MEME analysis, the maximum number of motifs was set to 15, with all other parameters set to default.

### 4.5. Cis-Elements Analysis of TaRboh Genes

To analyze the *cis*-elements of *TaRboh* genes, 1.5 kb upstream regions from the transcription start sites of each *TaRboh* gene were extracted from the wheat gene files. *Cis*-elements were predicted using PlantCARE website (http://bioinformatics.psb.ugent.be/webtools/plantcare/html/, accessed on 7 Decmeber 2023) [63], and results were visualized using R language.

### 4.6. Gene Duplication and Collinerity Analysis

Gene duplication events within the wheat genome were analyzed using the Multiple Collinearity Scan toolbox (MCScanX) program [64] with a BLASTp search (e-value < 10^−5^). Apart from wheat, we have chosen three monocots (*Brachypodium*, rice, and maize) and three dicots (*Glycine*, *Arabidopsis*, and *Aegilops*) to examine the evolutionary traits of *Rboh* genes in seven different species. Evolutionary relationships among wheat, *Aegilops*, *Brachypodium*, rice, maize, *Arabidopsis*, and *Glycine* were investigated using the TimeTree website [65]. Using the wheat genome as the reference, TBtools V2.136 software was employed to identify collinear gene pairs among these species. Gene duplication within species and interspecies gene collinearity relationships were visualized using TBtools V2.136 software. The non-synonymous (Ka) and synonymous (Ks) substitution rates for gene pairs were calculated with KaKs_Calculator 2.0 software [66]. Positive selection is indicated by a Ka/Ks ratio greater than 1, negative selection is shown by a Ka/Ks ratio less than 1, and neutral selection is indicated by a Ka/Ks ratio equal to 1. Divergence times of collinear gene pairs were estimated using the formula $T=Ks/(2\lambda \times 10^{-6})$ Mya, with λ set to $6.5\times 10^{-9}$.

### 4.7. Prediction of Protein–Protein Interactions Network and MicroRNA Targets

The STRING database (version 12.0) [67] (http://string-db.org/, accessed on 26 July 2023) was used to analyze protein–protein interactions among all TaRboh proteins with medium confidence (0.4). The interaction network was then visualized and enhanced using Cytoscape software (version 3.10.1) (http://www.cytoscape.org/) [68]. To predict potential miRNA binding sites for *TaRboh* genes, *TaRboh* cDNA sequences were submitted to the psRNATarget website with published miRNAs in wheat using default settings [69].

### 4.8. Expression Analysis of TaRboh Genes

RNA-seq data for wheat *TaRboh* genes under different stress conditions (drought, heat, and combined drought and heat) were obtained from the expVIP database [19]. The raw data were processed using $\log_{2}\left(TPM+1\right)$ transformation and visualized using the ‘HeatMap’ function in TBtools V2.136 software.

### 4.9. RNA Isolation and Quantitative Real-Time PCR (qRT-PCR)

Total RNA was extracted from plants subjected to drought, heat, and combined drought and heat stress at 1 h and 6 h using the Spin Column Plant Total RNA Purification Kit from Sangon Biotech. The extracted RNA was qualitatively assessed using electrophoresis and a NanoDrop 1000. First-strand cDNA synthesis was performed using NovoScript^®^ Plus All-in-One 1st Strand cDNA Synthesis SuperMix (gDNA Purge). Quantitative real-time PCR (qRT-PCR) was conducted using Hieff^®^ qPCR SYBR Green Master Mix (No Rox), with *TaRP15* as the internal reference gene. Nine representative *TaRboh* genes that strongly responded to heat and drought were validated using qRT-PCR investigation. The primers used for qRT-PCR are listed in Appendix A.

### 4.10. Statistical Analysis

The relative expression levels were calculated from triplicate biological and technical measurements using the $2^{-△△Ct}$ method, where $△△Ct={CT}_{target∕Cd}-{CT}_{actin∕Cd}-({CT}_{target∕control}-{CT}_{actin∕control})$. The information was displayed as mean ± standard error (S.E.). Statistical comparisons were conducted using one-way ANOVA, with differences considered significant at p<0.05. GraphPad Prism v9.5.0 was employed to visualize the obtained results.

## 5. Conclusions

Using bioinformatics analysis, 39 *TaRboh* genes were found in this study based on the most recent wheat genome assembly. The results from phylogenetic study show that *TaRboh*s were closely related to *OsRboh*s. *TaRboh*s were divided into ten groups based on their phylogenetic relationships, structures, and motifs. According to a collinearity analysis, *TaRboh*s had severe purifying selection during evolution, which led to the loss of certain wheat genes, in contrast to *Arabidopsis*. An investigation of protein–protein interactions revealed connections between the TaRboh proteins. According to miRNA target predictions, *TaRboh* gene regulation might be influenced by a number of miRNAs. Furthermore, *TaRboh* promoter predictions suggested that *TaRboh*s are important in plant growth and development, responses to biotic and abiotic stressors, and phytohormone signaling. As shown in the expression analysis, *TaRboh*s responded to heat, drought, and a combination heat and drought conditions, and these results were confirmed in the qPCR analysis. Results presented in this study form a solid foundation for future research into *TaRboh*-related stress resistance mechanisms in wheat.

## Figures and Tables

**Figure 1 plants-13-03377-f001:**
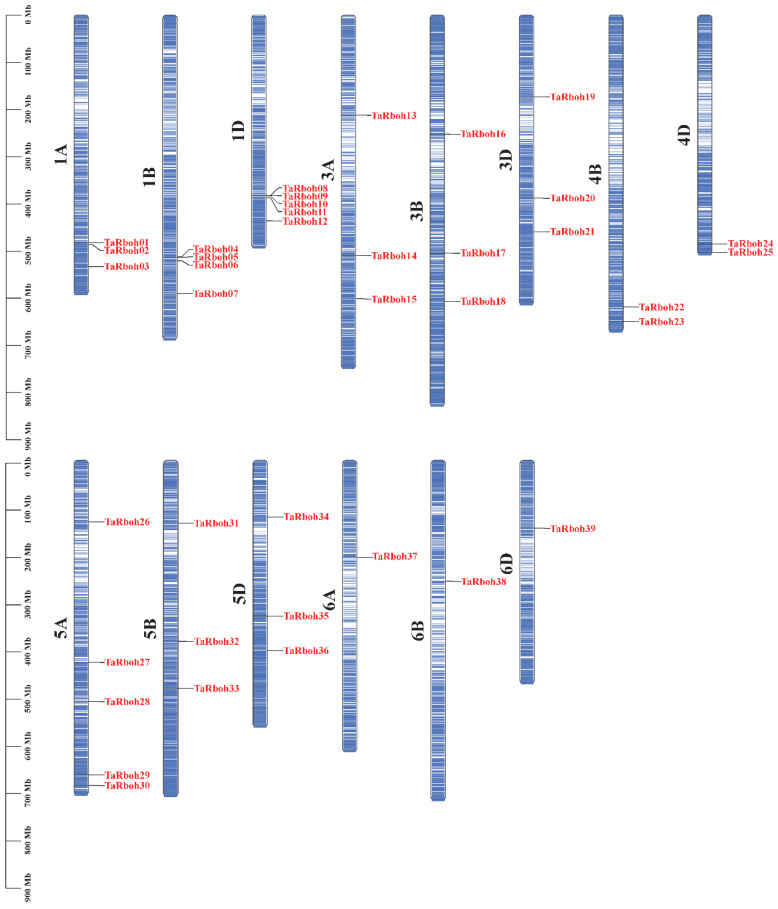
Locations of 39 *TaRboh* genes in the wheat genome. The color variation on the chromosomes represents gene density, ranging from low (white) to high (blue).

**Figure 2 plants-13-03377-f002:**
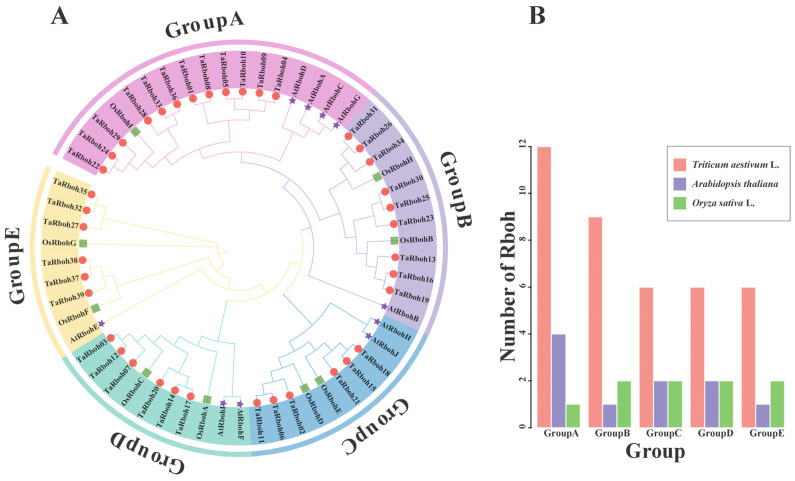
*Rboh* gene family in wheat, *Arabidopsis*, and rice. (**A**) The phylogenetic tree constructed for Rhoh sequences from wheat, *Arabidopsis*, and rice using MEGA-X based on the neighbor-joining (NJ) method with the p-distance substitution model (gamma = 1) and 1000 bootstrap replicates. The Rboh proteins of wheat are denoted by red circles. The Rboh proteins of rice are denoted by green squares. The Rboh proteins of *Arabidopsis* are denoted by purple pentagrams. (**B**) The number of *Rboh* genes belonging to different groups in wheat, *Arabidopsis*, and rice, respectively.

**Figure 3 plants-13-03377-f003:**
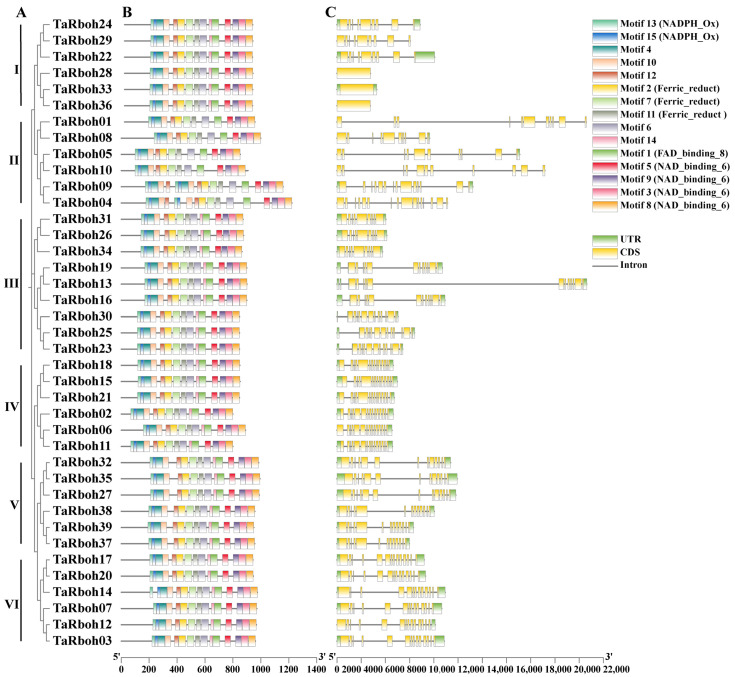
Evolutionary relationships, conserved motifs, and gene structures of *TaRboh* genes. (**A**) Phylogenetic tree of *TaRboh* genes constructed using MEGA-X with the Maximum Likelihood (ML) method. (**B**) Conserved motifs in TaRboh proteins, with fifteen distinct motifs identified. (**C**) Structures of *TaRboh* genes, where green rectangles represent untranslated regions (UTRs), yellow rectangles denote exons, and short lines indicate introns.

**Figure 4 plants-13-03377-f004:**
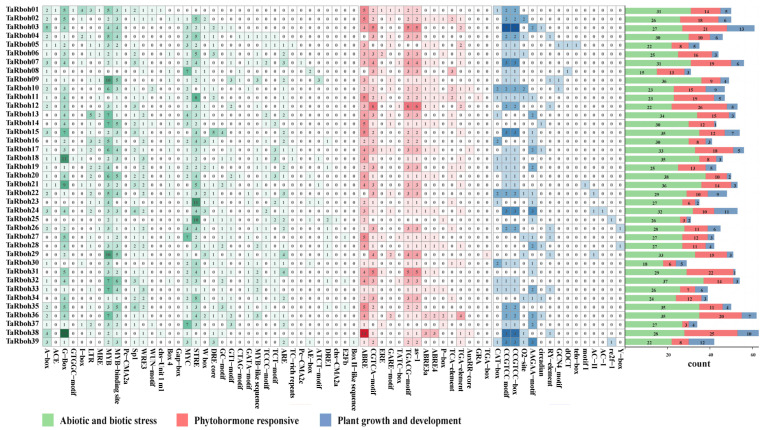
Distribution of *cis*-elements in the promoter regions of the *TaRboh* genes. The color and number of grids represent the quantity of *cis*-elements in each of the *TaRboh* genes.

**Figure 5 plants-13-03377-f005:**
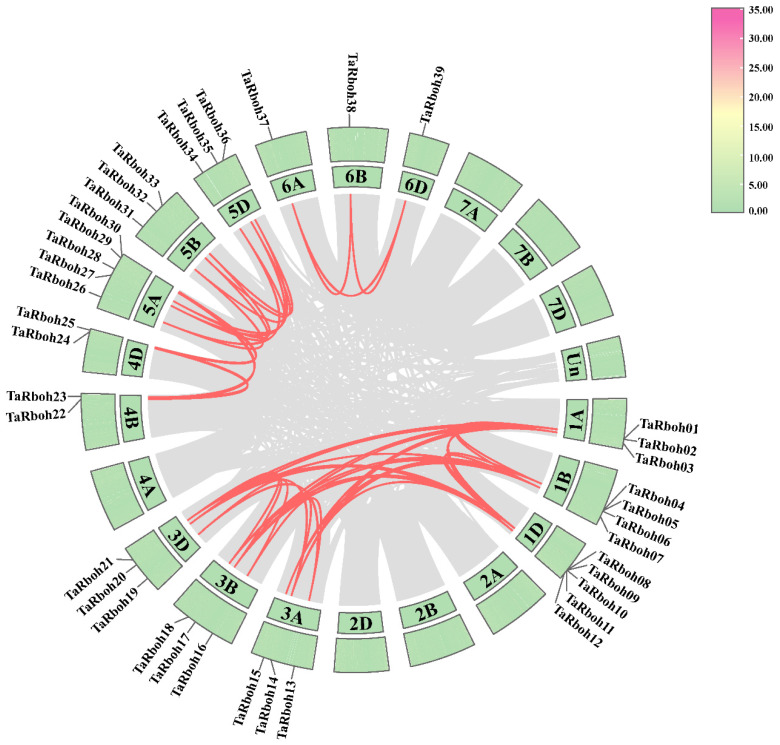
Collinearity analysis of *TaRboh* genes in wheat. Red lines indicate the segmentally duplicated *TaRboh* gene pairs in wheat. Gray lines indicate other gene pairs, excluding *TaRboh* genes. Chromosome lengths (Mb) are marked on the scale bar.

**Figure 6 plants-13-03377-f006:**
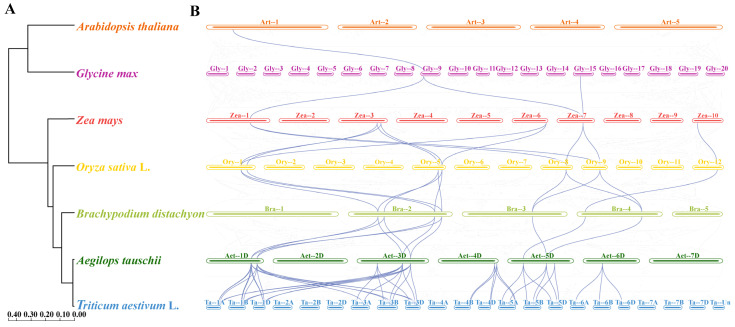
Synteny analysis of *TaRboh* genes between wheat and six other plant species (maize, rice, *Brachypodium*, *Glycine*, *Arabidopsis*, and *Aegilops*) based on the species’ evolutionary tree on the left. (**A**) Evolutionary relationships among the seven species predicted based on their whole genomes. Node numbers represent the estimated divergence times. (**B**) Collinearity relationships among the seven plant species based on the *TaRboh* genes in wheat. Gray lines in the background indicate syntenic blocks between adjacent species, while the blue lines highlight the linked genes identified.

**Figure 7 plants-13-03377-f007:**
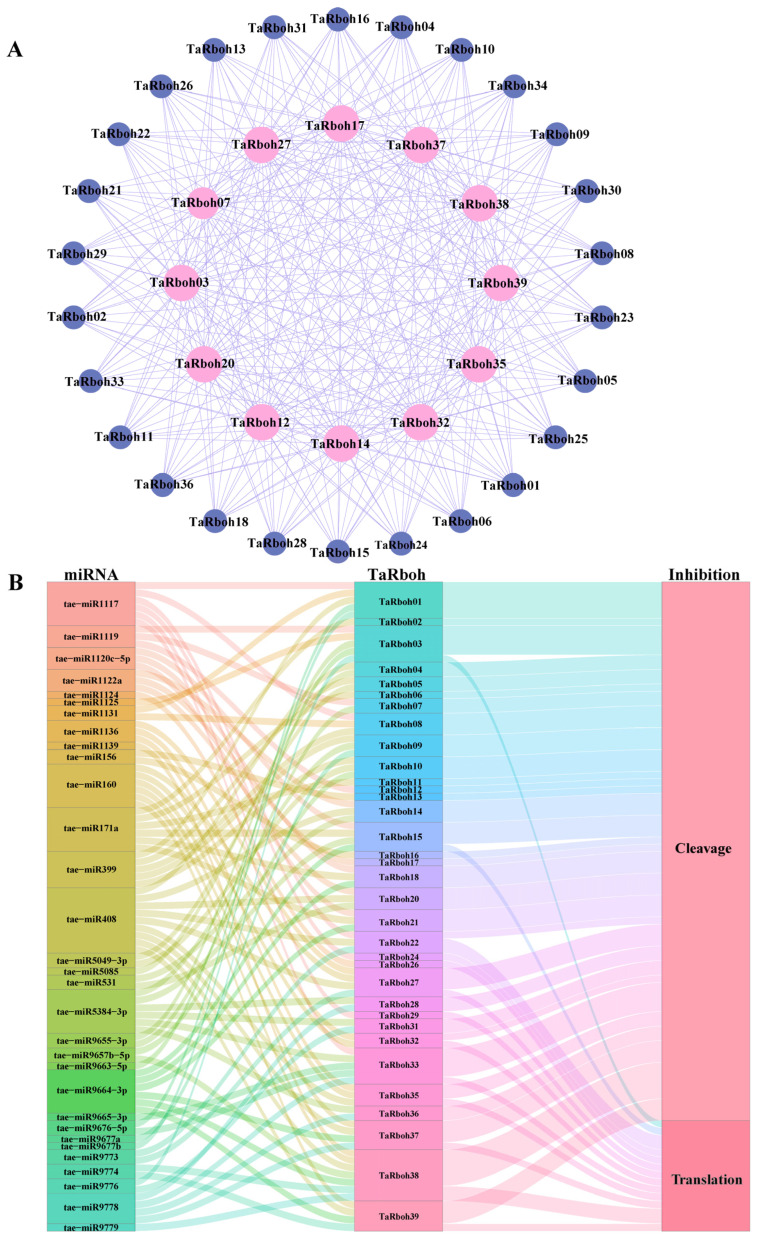
Protein–protein interaction network and miRNA target analysis of *TaRboh*s. (**A**) Red represents 13 TaRboh proteins with a high degree of 26. Blue represents 26 TaRboh proteins with a low degree of 12. (**B**) Sankey diagram depicting relationships of TaRboh transcripts targeted by miRNAs. The three columns represent miRNA, *TaRboh*, and the inhibition effect. Different colors are used to distinguish different mirnas, different *TaRboh*s, and different the inhibition effects.

**Figure 8 plants-13-03377-f008:**
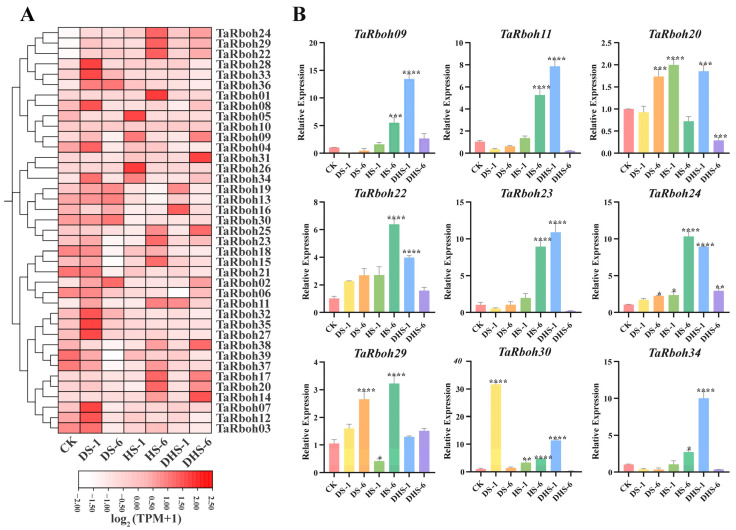
Expression profiles of *TaRboh* genes under different stress conditions. (**A**) Heatmap showing expression of *TaRboh*s under different treatments. Gene expression is expressed in $\log_{2}\left(TPM+1\right)$. Relative expression levels under three different abiotic stress conditions (control (CK), drought stress for 1 h (DS-1), drought stress for 6 h (DS-6), heat stress for 1 h (HS-1), heat stress for 6 h (HS-6), combined drought and heat stress for 1 h (DHS-1), combined drought and heat stress for 6 h (DHS-6)). CK: control. DS-1 (6): drought stress for 1 (6) h; HS-1 (6): heat stress for 1 (6) h; DHS-1 (6): combined drought and heat stress for 1 (6) h. (**B**) Quantitative expression of nine *TaRboh* genes in response to abiotic treatments including drought, heat and drought, and heat at 1 h and 6 h following treatments. The error bars indicate the standard deviations and the values in plots corresponding to the mean ± standard deviation (SD) of three independent biological replicates (* *p* < 0.05; ** *p* < 0.01; *** *p* < 0.001; **** *p* < 0.0001).

## Data Availability

Data are contained within the article and Appendix A.

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
