# Peer review of "Genome-Wide Identification, Classification, Expression Analysis, and Screening of Drought and Heat Resistance-Related Candidates of the Rboh Gene Family in Wheat"

_plants, 2024, doi:10.3390/plants13233377_

Round 1
Reviewer 1 Report
Comments and Suggestions for Authors
The Respiratory Burst Oxidase Homologs (RBOH) are key enzymes responsible for ROS production that play regulatory roles in many developmental processes as well as stress responses. Plants contain gene families of these oxidases, and it is thought that different oxidases participate in the regulation of distinct processes. Cao et al. laid the foundation for studying and characterizing the function of RBOH in wheat. Using bioinformatics analysis, they conducted a systematic investigation of the wheat Rboh gene family. They identified 39 genes containing all the important functional domains of these oxidases and performed comparative genomics, predicting the potential functions of different oxidases, with a particular focus on those involved in drought and heat stress.
This manuscript presents a thorough bioinformatics analysis of the TaRboh gene family in wheat, providing valuable insights into their potential roles in stress responses, particularly to drought and heat. This analysis yielded interesting results, such as the discovery that the number of Rboh genes in wheat is significantly higher than in other species, and this number does not correlate with genome size. Whole-genome duplication (WGD) is likely the most plausible explanation for this increase in the allohexaploid wheat genome. They also conducted phylogenetic tree analyses based on RBOH proteins, clustering the TaRboh genes, along with the Rboh of other plant species, into five distinct groups. Additionally, a collinearity analysis was performed to assess homology inference, revealing that five TaRboh genes are highly conserved throughout evolution and have significant research value. They analyzed the promoters and identified cis-regulatory elements mainly belonging to three functional groups: stress response, hormone response, developmental regulation, and biotic and abiotic stress responses. Furthermore, they predicted 31 distinct miRNAs that could target TaRboh genes in wheat.
Finally, they performed an expression/RNA-seq analysis using publicly available data and identified the TaRboh genes that differentially responded to heat, drought, and combined heat and drought conditions. In the only experimental work, they conducted a validation study to confirm the expression of some TaRboh genes through qRT-PCR analysis.
The analysis of the entire TaRboh gene family is important because it lays the foundation for characterizing the different Rboh genes related to various functions, particularly in response to abiotic stresses such as drought and heat, which are major factors limiting wheat production. The computational analyses they performed are interesting as they reveal important characteristics of the TaRboh genes.
However, the key information regarding the expression analysis, which could shed light on the functionality of the different TaRboh genes, is not consistent. Several assertions in the manuscript are not supported by the data. For instance, in the results section, line 265 states, “As shown in Figure 8B, expression patterns of these nine genes were consistent with the transcriptome data under different stress situations,” but this is not accurate. Only one gene (TaRboh11) of the nine analyzed shows expression patterns that align with the RNA-seq analyses. In the discussion, line 314 claims that transcriptomic analysis identified TaRboh03, TaRboh07, TaRboh12, TaRboh14, TaRboh17, and TaRboh20 as being related to drought stress, but in the figure, TaRboh14 and TaRboh20 do not seem to be induced by drought. Conversely, TaRboh27, TaRboh28, TaRboh32, TaRboh33, and TaRboh35 show greater induction than TaRboh03, TaRboh07, and TaRboh12, yet they are not mentioned in the discussion nor was their expression confirmed by qRT-PCR.
Also, in these expression analyses (particularly the RNA-seq), double stress seems to block the induced expression of some TaRboh genes observed under single stress conditions. The discussion should address this if the blockage is indeed confirmed. Aside from this, there are other minor issues:
In Figure 3, the motif analysis performed by TBtools is quite interesting, but it would be more informative to include the related functions (not just motif numbers), particularly since you mentioned some of these functions (lines 130-134). Would it be possible to add the assigned functions of the motifs to the figure list?
Line 192: If you write “the other six species,” you should list the species for the reader, as you did in the figure.
Line 215: Correct “miRNA” to have a lowercase “m” (miRNA).
Line 243: The stress condition "drought stress 1 hour" is not clearly stated. Organize the stress condicionas in a logical order.
Line 247: Separate “showeddistinct” into “showed distinct.”
In Figures 6 and 7, the lettering is very small in some cases, and in Figure 7b, it is almost unreadable. If necessary, consider moving some details to a supplementary file.
In Figure 8A, clearly detail all the treatments. Also, explain what the “(6)” appearing multiple times refers to.
Author Response
Reviewer #1: The Respiratory Burst Oxidase Homologs (RBOH) are key enzymes responsible for ROS production that play regulatory roles in many developmental processes as well as stress responses. Plants contain gene families of these oxidases, and it is thought that different oxidases participate in the regulation of distinct processes. Cao et al. laid the foundation for studying and characterizing the function of RBOH in wheat. Using bioinformatics analysis, they conducted a systematic investigation of the wheat Rboh gene family. They identified 39 genes containing all the important functional domains of these oxidases and performed comparative genomics, predicting the potential functions of different oxidases, with a particular focus on those involved in drought and heat stress.
This manuscript presents a thorough bioinformatics analysis of the TaRboh gene family in wheat, providing valuable insights into their potential roles in stress responses, particularly to drought and heat. This analysis yielded interesting results, such as the discovery that the number of Rboh genes in wheat is significantly higher than in other species, and this number does not correlate with genome size. Whole-genome duplication (WGD) is likely the most plausible explanation for this increase in the allohexaploid wheat genome. They also conducted phylogenetic tree analyses based on RBOH proteins, clustering the TaRboh genes, along with the Rboh of other plant species, into five distinct groups. Additionally, a collinearity analysis was performed to assess homology inference, revealing that five TaRboh genes are highly conserved throughout evolution and have significant research value. They analyzed the promoters and identified cis-regulatory elements mainly belonging to three functional groups: stress response, hormone response, developmental regulation, and biotic and abiotic stress responses. Furthermore, they predicted 31 distinct miRNAs that could target TaRboh genes in wheat.
Finally, they performed an expression/RNA-seq analysis using publicly available data and identified the TaRboh genes that differentially responded to heat, drought, and combined heat and drought conditions. In the only experimental work, they conducted a validation study to confirm the expression of some TaRboh genes through qRT-PCR analysis.
The analysis of the entire TaRboh gene family is important because it lays the foundation for characterizing the different Rboh genes related to various functions, particularly in response to abiotic stresses such as drought and heat, which are major factors limiting wheat production. The computational analyses they performed are interesting as they reveal important characteristics of the TaRboh genes.
However, the key information regarding the expression analysis, which could shed light on the functionality of the different TaRboh genes, is not consistent. Several assertions in the manuscript are not supported by the data.
Comments 1: For instance, in the results section, line 265 states, “As shown in Figure 8B, expression patterns of these nine genes were consistent with the transcriptome data under different stress situations,” but this is not accurate. Only one gene (TaRboh11) of the nine analyzed shows expression patterns that align with the RNA-seq analyses.
Response 1: Thank you for your suggestion . We have corrected it. Line 272 - Line 2822
Comments 2: In the discussion, line 314 claims that transcriptomic analysis identified TaRboh03, TaRboh07, TaRboh12, TaRboh14, TaRboh17, and TaRboh20 as being related to drought stress, but in the figure, TaRboh14 and TaRboh20 do not seem to be induced by drought. Conversely, TaRboh27, TaRboh28, TaRboh32, TaRboh33, and TaRboh35 show greater induction than TaRboh03, TaRboh07, and TaRboh12, yet they are not mentioned in the discussion nor was their expression confirmed by qRT-PCR.
Response 2: Thank you for your professional review. Transcriptome analysis showed that compared with the control group, TaRboh14 expression increased at 6 hours of drought stress (DS-6), while TaRboh20 expression was upregulated at 1 hour of drought stress (DS-1). These findings suggest a connection between drought stress and TaRboh14 and TaRboh20. Furthermore, we chose TaRboh03, TaRboh07, TaRboh12, TaRboh14, TaRboh17, and TaRboh20 for discussion in light of the phylogenetic tree (Figure 2) and the cis-elements (Figure 4). These genes all have drought stress-related cis-elements. Along with OsRbohA, they are also members of Group D, which has been linked to drought stress in the literature to date. Consequently, we deduce that TaRboh03, TaRboh07, TaRboh12, TaRboh14, TaRboh17, and TaRboh20 might also be connected to drought stress.
Comments 3: Also, in these expression analyses (particularly the RNA-seq), double stress seems to block the induced expression of some TaRboh genes observed under single stress conditions. The discussion should address this if the blockage is indeed confirmed.
Response 3: We have added content to the discussion. Line 387 - Line 393
Aside from this, there are other minor issues:
Comments 1: In Figure 3, the motif analysis performed by TBtools is quite interesting, but it would be more informative to include the related functions (not just motif numbers), particularly since you mentioned some of these functions (lines 130-134). Would it be possible to add the assigned functions of the motifs to the figure list?
Response 1: We have revised Figure 3.
Comments 2: Line 192: If you write “the other six species,” you should list the species for the reader, as you did in the figure.
Response 2: We have corrected it. Line 192
Comments 3: Line 215: Correct “miRNA” to have a lowercase “m” (miRNA).
Response 3: We have corrected it.
Comments 4: Line 243: The stress condition "drought stress 1 hour" is not clearly stated. Organize the stress condicionas in a logical order.
Response 4: We have revised the description of the stress conditions in a logical order.
Comments 5: Line 247: Separate “showeddistinct” into “showed distinct.”
Response 5: We have corrected it.
Comments 6: In Figures 6 and 7, the lettering is very small in some cases, and in Figure 7b, it is almost unreadable. If necessary, consider moving some details to a supplementary file.
Response 6: We have revised Figures 6 and 7.
Comments 7: In Figure 8A, clearly detail all the treatments. Also, explain what the “(6)” appearing multiple times refers to.
Response 7: ‘(6)’ refers to the stress treatment for 6 hours. We have revised the description of all treatments. Line 261 - Line 264

Reviewer 2 Report
Comments and Suggestions for Authors
In this manuscript, the authors have used bioinformatic and proteomic software tools to search for and analyze Rboh gene family members in wheat. They also have used qPCR to show changes in expression, in response to the heat and drought stress. While I feel this is a very good start to study the role of these genes in heat and drought stress, I feel the data may be too preliminary for publication and unlikely to be of real interest to readers of “plants”. It seems to me, many of the expression changes could be indirect, and since no attempt is made to test any suggestions, conclusions cannot be reached and, as indicated by the authors, the data are really only a foundation for future studies. Accordingly, I feel the manuscript is preliminary and should be rejected. A secondary concern which the authors may wish to address, is the fact that I feel all the software, etc. tools, which are used, are not well documented or referenced.In this manuscript.
Author Response
Reviewer #2: In this manuscript, the authors have used bioinformatic and proteomic software tools to search for and analyze Rboh gene family members in wheat. They also have used qPCR to show changes in expression, in response to the heat and drought stress.
Comments 1: While I feel this is a very good start to study the role of these genes in heat and drought stress, I feel the data may be too preliminary for publication and unlikely to be of real interest to readers of “plants”. It seems to me, many of the expression changes could be indirect, and since no attempt is made to test any suggestions, conclusions cannot be reached and, as indicated by the authors, the data are really only a foundation for future studies. Accordingly, I feel the manuscript is preliminary and should be rejected.
Response 1: Thank you for your professional review. This study systematically analyzed the wheat Rboh gene family using bioinformatics techniques. In addition, we selected 9 representative TaRboh genes and performed qPCR detection to validate the transcriptome results. This study provides a theoretical basis for further research on how the Rboh gene responds to high temperature and drought stress. Further experimental analysis is needed to confirm gene function and regulatory mechanisms.
Comments 2: A secondary concern which the authors may wish to address, is the fact that I feel all the software, etc. tools, which are used, are not well documented or referenced.In this manuscript.
Response 2: We have reviewed the content of the Materials and Methods section and found that there was an error in the citation of the transcriptomic data references, which has now been corrected. Line 571 - Line 573

Reviewer 3 Report
Comments and Suggestions for Authors
The manuscript is devoted to one of the topical problems – responses of important agricultural plants (wheat) to stress effects, in particular to the effect of high temperatures and lack of moisture. Identification of genes involved in responses to such stress effects, including the TaRboh genes, and identification of their mechanism of action, seems to be a topical problem of a fundamental nature, and is also of great practical importance in the creation of new plant varieties.
The manuscript can be recommended for publication in the journal, taking into account the comments below:
1. The authors should more clearly and in detail outline in the manuscript the relevance and novelty of their research in comparison with previously published works on wheat, both in the introduction and in the discussions.
Sharma, Y.; Ishu; Shumayla; Dixit, S.; Singh, K.; Upadhyay, S.K. Decoding the Features and Potential Roles of Respiratory Burst Oxidase Homologs in Bread Wheat. Curr. Plant Biol. 2024, 37, 100315, doi:10.1016/j.cpb.2023.100315.
Hu CH, Wei XY, Yuan B, Yao LB, Ma TT, Zhang PP, Wang X, Wang PQ, Liu WT, Li WQ, Meng LS, Chen KM. Genome-Wide Identification and Functional Analysis of NADPH Oxidase Family Genes in Wheat During Development and Environmental Stress Responses. Front Plant Sci. 2018 Jul 23;9:906. doi: 10.3389/fpls.2018.00906.
2. RT-PCR analysis Statistical processing of data was performed incorrectly. Student's t test is only intended to compare two independent groups when the necessary conditions for the use of parametric criteria. There are three groups in your work, for example: control, drought stress for 1 h, drought stress for 6 h.
Comparison three or more independent groups quantitative data is carried outusing one-dimensional (one-factor) disperson analysis (One-Way ANOVA)
or Kruskal-Wallis test (Kruskal-Wallis test). Kruskal-Wallis test will help
you find out if there are differences between groups, but will not be able to
show between which groups these differences exist. When statistically significant
differences are found between groups using the Kruskal-Wallis test further
a posteriori comparison should be made using the Mann-Whitney test. 3. Fig.8. You must add to your signature: The error bars indicate the standard deviations and the values in plots
corresponding to the mean ± standard deviation (SD) of three independent
biological replicates (* p < 0.05; ** p < 0.01; *** p < 0.001).
Author Response
Reviewer #3: The manuscript is devoted to one of the topical problems – responses of important agricultural plants (wheat) to stress effects, in particular to the effect of high temperatures and lack of moisture. Identification of genes involved in responses to such stress effects, including the TaRboh genes, and identification of their mechanism of action, seems to be a topical problem of a fundamental nature, and is also of great practical importance in the creation of new plant varieties.
The manuscript can be recommended for publication in the journal, taking into account the comments below:
Comments 1: The authors should more clearly and in detail outline in the manuscript the relevance and novelty of their research in comparison with previously published works on wheat, both in the introduction and in the discussions.
Sharma, Y.; Ishu; Shumayla; Dixit, S.; Singh, K.; Upadhyay, S.K. Decoding the Features and Potential Roles of Respiratory Burst Oxidase Homologs in Bread Wheat. Curr. Plant Biol. 2024, 37, 100315, doi:10.1016/j.cpb.2023.100315.
Hu CH, Wei XY, Yuan B, Yao LB, Ma TT, Zhang PP, Wang X, Wang PQ, Liu WT, Li WQ, Meng LS, Chen KM. Genome-Wide Identification and Functional Analysis of NADPH Oxidase Family Genes in Wheat During Development and Environmental Stress Responses. Front Plant Sci. 2018 Jul 23;9:906. doi: 10.3389/fpls.2018.00906.
Response 1: Thank you for your suggestion. We have discussed the above two studies in both the introduction and discussion sections. Line 67 - Line 70 and Line 288 - Line 293
Comments 2: RT-PCR analysis Statistical processing of data was performed incorrectly. Student's t test is only intended to compare two independent groups when the necessary conditions for the use of parametric criteria. There are three groups in your work, for example: control, drought stress for 1 h, drought stress for 6 h. Comparison three or more independent groups quantitative data is carried out using one-dimensional (one-factor) disperson analysis (One-Way ANOVA) or Kruskal-Wallis test (Kruskal-Wallis test). Kruskal-Wallis test will help
you find out if there are differences between groups, but will not be able to show between which groups these differences exist. When statistically significant differences are found between groups using the Kruskal-Wallis test further a posteriori comparison should be made using the Mann-Whitney test.
Response 2: We have corrected the errors in the experimental method. Line 490
Comments 3: Fig.8. You must add to your signature: The error bars indicate the standard deviations and the values in plots corresponding to the mean ± standard deviation (SD) of three independent
biological replicates (* p < 0.05; ** p < 0.01; *** p < 0.001).
Response 3: We have corrected it. Line 267 - Line 269

Round 2
Reviewer 1 Report
Comments and Suggestions for Authors
The authors have responded to the reviewers' comments and have made several corrections to the manuscript, addressing many of the requested revisions. However, the primary issue raised—concerning the discrepancy between the RNA-seq data and the qRT-PCR validation for various TaRboh genes under drought, heat, and combined stress conditions—remains inadequately addressed. The authors have made only minor modifications to their commentary on this issue (see Lines 272-282), without fully reconciling the inconsistencies in expression patterns observed between these two methods. This issue is critical, as the differential expression analysis under these specific stress conditions is central to attributing functional roles to individual TaRboh genes within stress response pathways. Without reliable qRT-PCR validation of the RNA-seq data, the results cannot robustly support the functional assignments proposed for these gene family members. As it stands, the RNA-seq findings lack the validation necessary to confirm their relevance in identifying specific gene functions under stress, which is essential for interpreting the data accurately and ensuring the conclusions are well-founded.
Author Response
Reviewer #1: The authors have responded to the reviewers' comments and have made several corrections to the manuscript, addressing many of the requested revisions. However, the primary issue raised—concerning the discrepancy between the RNA-seq data and the qRT-PCR validation for various TaRboh genes under drought, heat, and combined stress conditions—remains inadequately addressed. The authors have made only minor modifications to their commentary on this issue (see Lines 272-282), without fully reconciling the inconsistencies in expression patterns observed between these two methods. This issue is critical, as the differential expression analysis under these specific stress conditions is central to attributing functional roles to individual TaRboh genes within stress response pathways. Without reliable qRT-PCR validation of the RNA-seq data, the results cannot robustly support the functional assignments proposed for these gene family members. As it stands, the RNA-seq findings lack the validation necessary to confirm their relevance in identifying specific gene functions under stress, which is essential for interpreting the data accurately and ensuring the conclusions are well-founded.
Response 1: Thank you for your professional review. We conducted a literature search and found RNA-seq data for wheat under drought, heat-drought, and heat stress conditions. After preliminary analysis, we observed that TaRboh genes exhibit stress-specific expression under these three conditions. We selected 9 representative TaRboh genes that showed significant expression under one or more stresses to further confirm this result. We conducted qPCR experiments based on the information provided in the references.
The transcriptome data and experimental procedures were referenced from the following study: Liu, Z.; Xin, M.; Qin, J.; Peng, H.; Ni, Z.; Yao, Y.; Sun, Q. Temporal Transcriptome Profiling Reveals Expression Partitioning of Homeologous Genes Contributing to Heat and Drought Acclimation in Wheat (Triticum Aestivum L.). BMC Plant Biol 2015, 15, 152, doi:10.1186/s12870-015-0511-8.

Reviewer 2 Report
Comments and Suggestions for Authors
No additional comments.
Author Response
Reviewer #2: No additional comments.
Response: Thank you for your valuable comments, which have greatly enhanced the readability and scientific quality of the revised manuscript.

Round 3
Reviewer 1 Report
Comments and Suggestions for Authors
Thank you for the additional context and efforts to address the review comments. However, the primary concern about the discrepancy between RNA-seq and qRT-PCR validation for TaRboh gene expression remains unresolved. As noted, the validation results for 8 out of the 9 selected TaRboh genes diverged significantly from the RNA-seq data under the two stress conditions and the combination of both. This inconsistency raises substantial concerns regarding the reliability of the RNA-seq findings for accurately determining the stress-specific expression profiles of these genes. So, without a clear alignment between RNA-seq and qRT-PCR data, it is challenging to ascertain which genes are genuinely responsive to drought, heat, or combined stress, and thus difficult to attribute functional roles to individual TaRboh family members. Given the importance of accurate differential expression data for interpreting the roles of these genes in stress pathways, the unresolved discrepancies raises an important doubt on the manuscript’s key conclusions. Further investigation or clarification is needed to reconcile these differences and ensure the robust data is published.
Author Response
Reviewer #1: Thank you for the additional context and efforts to address the review comments. However, the primary concern about the discrepancy between RNA-seq and qRT-PCR validation for TaRboh gene expression remains unresolved. As noted, the validation results for 8 out of the 9 selected TaRboh genes diverged significantly from the RNA-seq data under the two stress conditions and the combination of both. This inconsistency raises substantial concerns regarding the reliability of the RNA-seq findings for accurately determining the stress-specific expression profiles of these genes. So, without a clear alignment between RNA-seq and qRT-PCR data, it is challenging to ascertain which genes are genuinely responsive to drought, heat, or combined stress, and thus difficult to attribute functional roles to individual TaRboh family members. Given the importance of accurate differential expression data for interpreting the roles of these genes in stress pathways, the unresolved discrepancies raises an important doubt on the manuscript’s key conclusions. Further investigation or clarification is needed to reconcile these differences and ensure the robust data is published.
Response: Thank you very much for your careful review and helpful suggestions. We conducted multiple repeated experiments and reanalyzed the experimental data. After that, The qPCR results of the five (TaRboh09, TaRboh22, TaRboh24, TaRboh30, and TaRboh34) of the eight genes are consistent with the RNA seq results. In fact, RNA seq, as a high-throughput research method, has a certain degree of bias in reflecting gene expression. The updated data is shown in Figure 8B, and we have made corresponding revisions to the manuscript. Line 26 - Line 28 and Line 273 - Line 281
